# A scalable membrane electrode assembly architecture for efficient electrochemical conversion of $CO_2$ to formic acid

Leiming Hu [1], Jacob A. Wrubel[1], Carlos M. Baez-Cotto [2], Fry Intia[1], Jae Hyung Park[3], Arthur Jeremy Kropf [3], Nancy Kariuki[3], Zhe Huang[4], Ahmed Farghaly [3], Lynda Amichi[5], Prantik Saha[1], Ling Tao [4], David A. Cullen [5], Deborah J. Myers [3], Magali S. Ferrandon[3] & K. C. Neyerlin [1] ✉

The electrochemical reduction of carbon dioxide to formic acid is a promising pathway to improve $CO_2$ utilization and has potential applications as a hydrogen storage medium. In this work, a zero-gap membrane electrode assembly architecture is developed for the direct electrochemical synthesis of formic acid from carbon dioxide. The key technological advancement is a perforated cation exchange membrane, which, when utilized in a forward bias bipolar membrane configuration, allows formic acid generated at the membrane interface to exit through the anode flow field at concentrations up to 0.25 M. Having no additional interlayer components between the anode and cathode this concept is positioned to leverage currently available materials and stack designs ubiquitous in fuel cell and $H_2$ electrolysis, enabling a more rapid transition to scale and commercialization. The perforated cation exchange membrane configuration can achieve >75% Faradaic efficiency to formic acid at <2 V and 300 mA/cm² in a 25 cm² cell. More critically, a 55-hour stability test at 200 mA/cm² shows stable Faradaic efficiency and cell voltage. Technoeconomic analysis is utilized to illustrate a path towards achieving cost parity with current formic acid production methods.

The electrochemical reduction of carbon dioxide to formic acid using renewable electricity has been shown to reduce production cost compared to traditional fossil-based approaches by as much as 75%[1]. As noted throughout the literature[2,3], formic acid has a wide range of applications, from an effective and economical hydrogen-storage and transportation medium, to a feedstock for the chemical[4,5] or biomass industry[6]. Formic acid has even been identified as an input for subsequent conversion to sustainable jet fuel intermediates utilizing metabolic engineering[7,8]. With momentum building for a formic acid economy[1,9], multiple research efforts have focused on the optimization of catalyst selectivity[10–16]. However, many efforts still focus on small-scale H-cell or liquid flow cells operating at low current densities (<50 mA/cm²). To reduce cost, enable commercialization, and increase subsequent market penetration, electrochemical carbon dioxide reduction ($CO_2$R) must be performed at high current densities (≥200 mA/cm²) and Faradaic efficiency (FE)[17] while maximizing the utilization of materials and stack components from fuel cell and water electrolysis technology that enable $CO_2$R devices to leverage

[1]Chemistry and Nanoscience Center, National Renewable Energy Laboratory, Golden, CO, USA. [2]Materials Science Center, National Renewable Energy Laboratory, Golden, CO, USA. [3]Chemical Sciences and Engineering Division, Argonne National Laboratory, Lemont, IL, USA. [4]Catalytic Carbon Transformation & Scale-Up Center, National Renewable Energy Laboratory, Golden, CO, USA. [5]Center for Nanophase Materials Sciences, Oak Ridge National Laboratory, Oak Ridge, TN, USA. ✉e-mail: Kenneth.Neyerlin@nrel.gov

economies of scale[18]. Furthermore, to enhance the production utility and avoid additional downstream processing, formic acid should be targeted as an end product instead of formate salt[19].

Along these lines, recent efforts have been made to develop industrially relevant gas diffusion electrode (GDE) based $CO_2R$ to formate/formic acid devices. Fernández-Caso et al.[20] made a comprehensive review summarizing all the electrochemical cell configurations for continuous reduction of $CO_2$ to formic acid/formate. In general, all existing configurations can be categorized into three main groups: 1. Flowing catholyte[19,21–27], 2. Single membrane (either cation exchange membrane (CEM)[28] or anion exchange membrane (AEM))[29], and 3. Interlayer configurations[15,30–32]. Simplified cross-sections for these configurations are shown in Fig. 1a. For flowing catholyte configurations, an electrolyte chamber is created between the membrane and the cathode GDE. The flowing catholyte serves to provide ionic access to the cathode catalyst layer[33], though its necessity to control formate selectivity has been debated[34]. Nevertheless, using such a configuration Chen et al. achieved up to 90% FE to formate at 500 mA/cm² using a carbon-supported $SnO_2$ cathode with 1.27 mm thick catholyte layer[35]. The thick catholyte layer, coupled with a reverse-bias bipolar membrane (BPM) to limit ion crossover, resulted in an operating voltage of 6 V and a 15% energy efficiency. To improve energy efficiency, Lee et al. used a single CEM configuration, achieving 93.3% FE at a partial current density of 51.7 mA/cm² [29]. Díaz-Sainz et al.[28] used a filter press setup with a single CEM membrane and can achieve 89% FE at a current density of 45 mA/cm². However, all approaches yielded formate as opposed to the preferred product, formic acid. Additional processing requirements aside, in CEM configurations, formate salt (e.g. KCOOH)

rapidly accumulates in the GDE and flow field resulting in transport limitations and eventual cell failure.

To combat the formation of formate salt, Proietto et al.[32] used an undivided filter-press cell configuration, with DI water flowing through the interlayer. The system can achieve >70% FE in the current density range of 50-80 mA/cm². In a similar vein, Yang et al.[14] introduced the use of a solid electrolyte interlayer between the CEM and AEM to promote the formation of formic acid. Yang et al.[31,36] achieved 91.3% FE at 200 mA/cm² in a 5 cm² cell, yielding a 6.35 wt.% formic acid solution. Xia et al. used a similar configuration and achieved 83% carbon dioxide ($CO_2$) to formic acid FE at 200 mA/cm², examining the durability of the system over 100 hrs[30]. While small-scale results are promising, the added cost and complexity of the porous ion-exchange resin make interlayer configurations challenging to scale to larger systems (e.g. 1000 cm²).

To help visualize the net impact of the various designs, we tabulated formate/formic acid production per kWh for all the previously referenced systems and plotted them in Fig. 1b. Here, it is evident that any system containing a catholyte or interlayer reaches a maximum in performance at low current densities with performance decreasing at higher current densities, where ohmic limitations can dictate cell voltage. Additionally, while energy-efficient CEM configurations yield the highest production of $mol_{formic\ acid}$/kWh, salt accumulation results in a rapid decrease in performance at high current densities.

To mitigate the previously discussed failure modes, we've developed a membrane electrode assembly (MEA) containing a composite forward bias BPM with a perforated cation exchange membrane (PCEM). This architecture is detailed in Fig. 1c. The anode is supplied

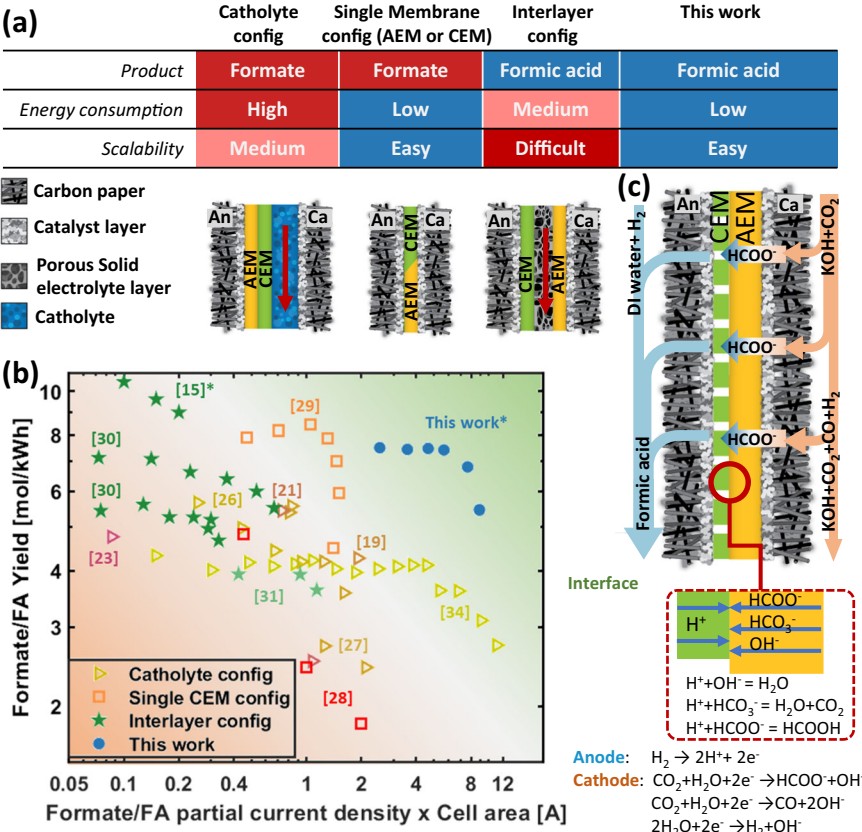

**Fig. 1 | Comparison of different CO2R to formate/formic acid configurations.**
**a** Comparison of the three most prominent device configurations for CO2R to formate/formic acid, along with the architecture proposed in this study.
**b** Comparison of total current and formate/formic acid yield for catholyte configuration, interlayer configuration, single CEM configuration from literature as shown in supplemental Table S1 and our work. Hollow markers represent the production of formate salt solution, while solid markers indicate the production of formic acid. * Stands for configurations using hydrogen at the anode. **c** The structure of zero-gap MEA configuration using composite bipolar membrane with perforated cation exchange layer operating in forward-bias mode.

with hydrogen ($H_2$), and protons are generated by hydrogen oxidation reaction (HOR). The introduction of a PCEM layer in a BPM system enables formate ions generated at the cathode to traverse the AEM, combine with protons to form formic acid at the BPM interface and interstitial CEM pores, subsequently departing through the anode GDE and flow field. Using this configuration, we achieve >75% FE to formic acid at <2 V and 300 mA/cm$^2$ for a 25 cm$^2$ cell. Most critically, this design leverages commercially available components and device architectures used for fuel cell and water electrolyzer stacks, enabling a more rapid transition to scale. Catholyte configurations contain a catholyte flow chamber, which can lead to pressure imbalance between the gas and liquid phase especially at larger cell configurations. For interlayer configurations with a porous liquid flow layer, significant efforts are needed to optimize the porous interlayer to alleviate pressure drop and carbon dioxide build-up within the interlayer. Both of which can lead to cell failure. It is also difficult to fabricate a stand-alone thin, porous interlayer at a large scale. In contrast, the proposed new configuration is a zero-gap MEA configuration that does not contain any liquid flow chamber or interlayer. The uniqueness of the proposed configuration compared to other existing electrochemical cells is that it can directly synthesize formic acid in a scalable, energy-efficient zero-gap configuration.

## Results and Discussion
### Screening of zero-gap MEA configurations
To suppress $H_2$ evolution, a plethora of $CO_2$ reduction efforts have utilized MEA configurations and AEM membranes, coupled with high molarity electrolytes (e.g. 1–10 M KOH) to create alkaline conditions on the cathode (as shown in Fig. 2a). In these configurations, formate ions generated at the cathode traverse the membrane as negatively charged species where they then form KCOOH and are ushered out of the system through the anode KOH stream. While formate FE and cell voltage are initially favorable, Fig. 2b, stability tests resulted in an approximately 30% FE reduction over just 10 hrs (Fig. S1a–c). It should be noted that the use of 1 M KOH anolyte is critical to both minimize anode overpotential in basic oxygen evolution reaction (OER)

systems[37] and enable ionic accessibility within the cathode catalyst layer[33]. When anolyte concentration is reduced to 0.1 M KOH, both cell voltage and formate oxidation (formate loss) increase (Fig. S1d), illustrating the zero-sum tradeoff. The amount of formate oxidation is estimated based on the total mass balance, with details in the method section. The performance of using a MEA configuration and single CEM membrane is also investigated, with results shown in Fig. S1f, g. The formate FE collected from the cathode is >60% at 200 mA/cm$^2$ at the beginning of the test, but suffers from fast degradation within two hours due to cathode salt accumulation as previously discussed (Fig. S11).

To target formic acid generation, $H_2$ was supplied at the anode to a carbon-supported Pt (Pt/C) catalyst. As shown in Fig. 2d, forward bias BPMs that generate protons at the anode have been previously examined to enable formic acid production. The BPM configuration cell failed after 40 mins at 200 mA/cm$^2$, accompanied by a voltage surge to over 5 V (Fig. 2e). Significant delamination at the CEM/AEM interface was observed after the test. In addition to formate, anions such as carbonate, bicarbonate, and hydroxide can also transport through the AEM membrane, reacting with protons at the CEM/AEM interface, forming $CO_2$ gas and liquid water, which can lead to BPM delamination (Fig. 2f) and ultimately cell failure.

### Performance of MEA with PCEM/AEM configuration
Based on the performance and failure mechanisms of the above-mentioned configurations, a new MEA architecture was proposed as shown in Fig. 1c and detailed further in Fig. 3a[38]. Here, the PCEM layer provides pathways for formic acid and anions to migrate from the CEM/AEM interface, reducing species accumulation. Simultaneously, the PCEM interstitial pathways direct formic acid into the diffusion media and flow field, reducing the likelihood for formic acid oxidation. Polarization results using 80, 40 and 25 mm thick AEMs are shown in Fig. 3b. While the total cell voltage increases with AEM the thickness, as expected, use of thicker AEMs prevents formic acid back diffusion, increasing cathode pH and reducing $H_2$ production (Fig. 3c–e).

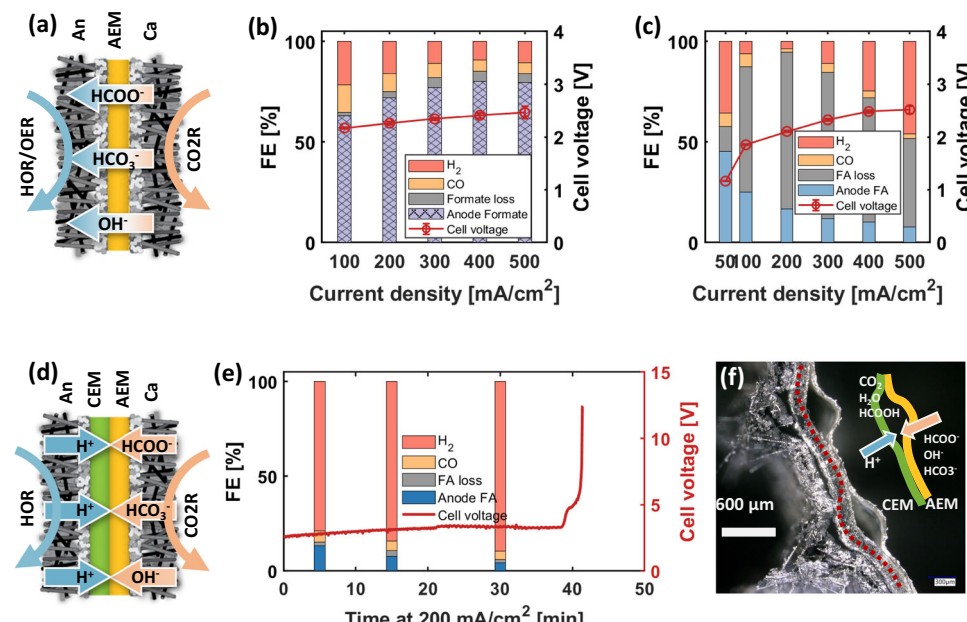

**Fig. 2 | Performance of two different zero-gap MEA configurations. a** Schematic of the zero-gap MEA with CO2R at the cathode with the hydrogen oxidation reaction (HOR) or OER at the anode, using a single AEM membrane in the middle. **b** FE and cell voltage of this configuration with flowing 1 M KOH and OER at the anode. Error bars stand for S.D. from three different measurements. **c** FE and cell voltage of the system with $H_2$ and HOR at the anode. Different colors are used to differentiate formate and formic acid production. **d** Schematic of MEA with forward-bias BPM in the middle. **e** FE and cell voltage vs. time at 200 mA/cm$^2$ using this configuration. **f** Cross-sectional image of the MEA with forward-bias BPM after short test.

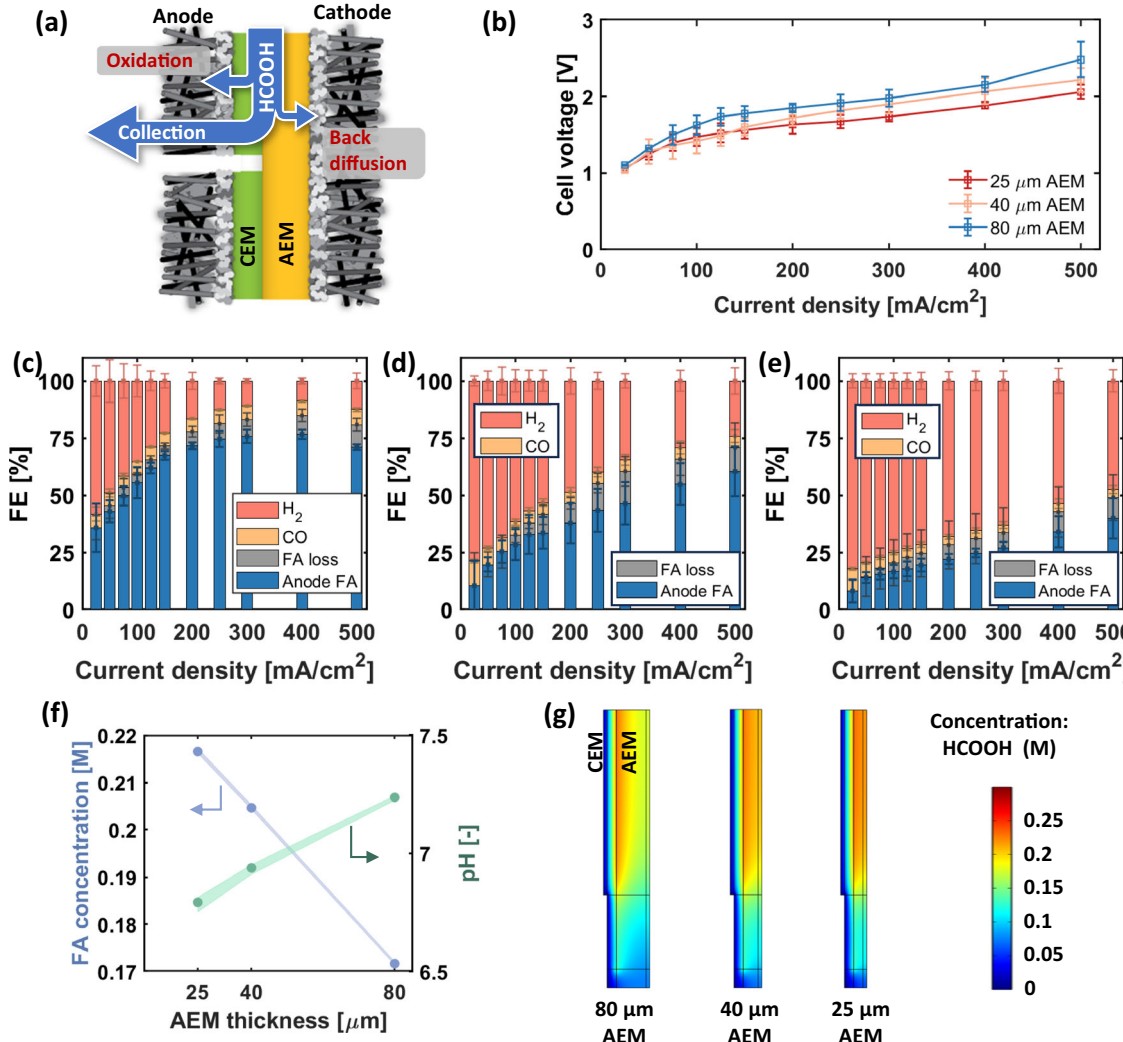

**Fig. 3 | Beginning of life performance of MEA using perforated CEM and AEM with different thicknesses. a** Illustration of the MEA structure with AEM and perforated CEM, and different formic acid transport pathways. **b** Cell voltage at different current densities with different AEM thicknesses. **c** FE at different current densities when the AEM thickness is 80 μm (**d**) 40 μm, (**e**) 25 μm. Error bars stand for S.D. measured from three individual samples. **f** Simulation results of formic acid concentration and pH across the CEM/AEM interface with different AEM thicknesses. **f** FA and pH in the cathode catalyst layer with different AEM membrane thicknesses. **g** 2D distribution of the formic acid concentration with CEM/AEM interface and perforation.

Figure S2 shows the formic acid concentration and pH distribution throughout the thickness direction of the MEA using finite element Poisson-Nernst-Plank simulation. It is not surprising that the CEM/AEM interface exhibits the greatest concentration of formic acid, 0.23 mol/L, as formic acid is produced at this interface. As the AEM thickness increases, there is a faster drop-off of formic acid concentration through the AEM, which suggests a greater mass transport resistance and a smaller flux of formic acid due to back diffusion. Figure 3f and g show the resulting pH and formic acid within the cathode catalyst layer caused by back diffusion, and 2D formic acid concentration distribution, respectively. With thinner AEM membranes, the formic acid concentration near the cathode is higher and cathode pH becomes acidic. Therefore, while thicker AEM membranes lead to higher ohmic losses, they are critical to prevent formic acid back diffusion to the cathode and maximizing a high net system formic acid FE. Ultimately, increasing AEM thickness to 80 μm enabled >75% FE to formic acid at <2 V and 300 mA/cm² for a 25 cm² cell.

### Stability of MEA with PCEM/AEM configuration
To test the stability of this PECM-based architecture, the cell was held at 200 mA/cm² for 55 hrs. The overall results are displayed in Fig. 4 with results from the first 3 hrs highlighted in Figure S3. When a Pt/C anode catalyst was used, the cell voltage increased dramatically within the first 30 minutes (Figure S3a). At longer durations, the cell voltage remained nearly constant, yielding a degradation rate of 0.6 mV/hr (Fig. 4a). At the beginning of test, the FE for formic acid collected at the anode is 76.5%, and the cathode FE for hydrogen is 19.2%. After the first hour of testing, the FE for hydrogen dropped to 13.8%, suggesting an improved selectivity for formate. However, the system FE for formic acid dropped to 62.7% at 1 hour, and the anode formic acid oxidation rate increased from near zero at the beginning of the test to 17.0%. Subsequently, the FE for H₂, CO, formic acid, and the anode formic acid oxidation rate remained stable for the duration of the experiments. The increase of formic acid oxidation over the first hour is likely related to formic acid accumulation at the PCEM/AEM interface. As the concentration of formic acid builds up, it will not only exit through membrane perforations but also via diffusion through the CEM itself, entering the Pt/C anode layer. Since formic acid is a liquid at 60 °C, its accumulation can cause mass transport issues and lead to preferential oxidation over hydrogen gas.

The morphology for both beginning-of-test (BOT), as prepared, and end-of-test (EOT), post 55-hr stability tested samples were

characterized using nano-Xray computed tomography (nano-CT), as shown in Fig. 5a. The EOT sample has a larger catalyst particle size, 1207 nm in diameter compared to 930 nm at BOT. The High-angle annular dark-field scanning transmission electron microscopy

(HAADF-STEM) image and energy-dispersive X-ray spectroscopy (EDS) results are shown in Fig. 5b. While the BOT catalyst layer contains a large portion of smaller catalyst particles, as well as some larger agglomerates, at the EOT, the catalyst layer can be divided into two separate regions: one with significantly larger solid particles and a more porous region containing a significant number of smaller particles. The EDS mapping shows that the large solid particles are Bi-rich, likely metallic Bi, while the porous region is oxygen-rich. When the cell is operated at 200 mA/cm², the negative potential at the cathode will lead to the reduction of $Bi_2O_3$, as evidenced by in situ X-ray absorption spectroscopy results discussed below. The HAADF-STEM and EDS mapping results indicate that $Bi_2O_3$ undergoes a reduction process, causing them to lose oxygen and coalesce into larger metallic particles. X-ray diffraction (XRD) patterns of the BOT and EOT cathodes support the interpretation of the EDS data (Fig. 5c): with only crystalline $Bi_2O_3$ detected in the BOT cathode and crystalline Bi-metal detected at the EOT. To understand the effect of cathode potential on the oxidation state of the $Bi_2O_3$ cathode catalyst, in-situ X-ray absorption spectra (XAS) were acquired at the Bi $L_3$ absorption edge in 0.1 M KOH from the open circuit potential (+0.3 V vs RHE) to −1.5 V vs RHE. The onset of the reduction of the $Bi_2O_3$ phase at −0.85 V vs RHE was observed, as indicated by a decrease in the white line intensity in the near-edge region of the spectrum, with 90% reduction to Bi metal at −1.1 V vs RHE (Fig. 5d). Regardless of the mechanism and despite the pronounced change in cathode morphology, catalyst oxidation state and crystallite structure, overall formate selectivity in the cathode, inferred from $H_2$ and CO FE along with formic acid production, remains largely unaffected.

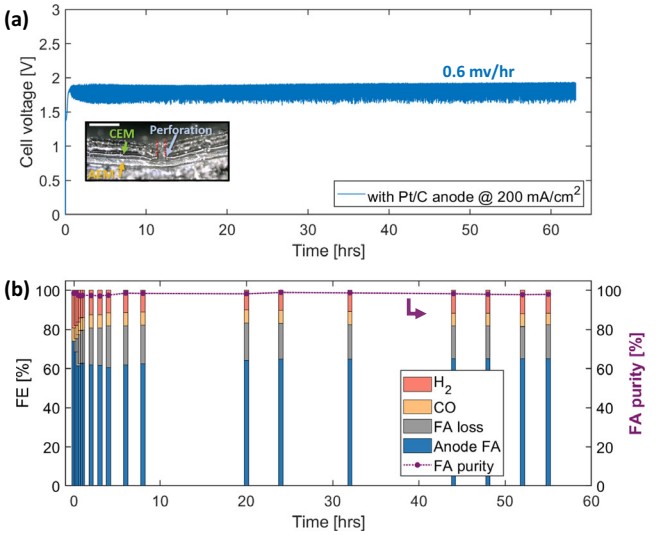

**Fig. 4 | Durability of the MEA with an 80 mm AEM and perforated CEM with a Pt/C anode. a** Cell voltage vs. time at 200 mA/cm² at 60 °C. Inset figure is an optical microscope image of the EOT cross-section of the MEA with perforated CEM. Scale bar: 300 μm. **b** FE and formic acid purity vs time at 200 mA/cm² using Pt/C anode.

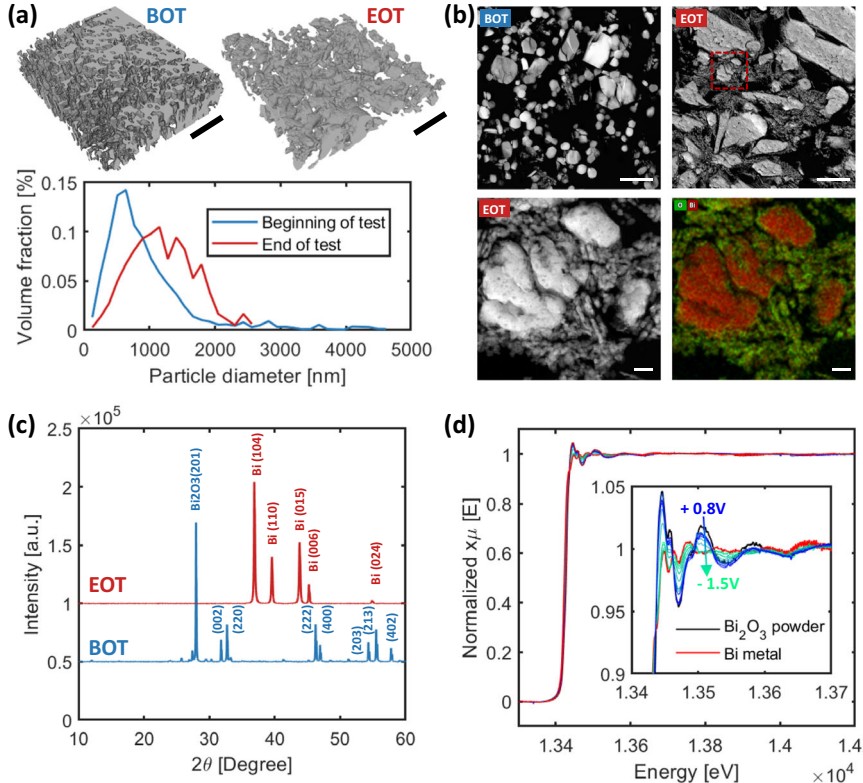

**Fig. 5 | Characterization results for beginning of test and end of test (55-hours stability tested) $Bi_2O_3$ cathode samples and results of in-situ X-ray spectroscopy studies of $Bi_2O_3$. a** 3D structures of the catalyst layer by nano-Xray CT, and the catalyst particle distributions. Scale bar: 10 μm. **b** Top 2: HAADF-STEM images of the BOT and EOT cathode catalyst layer. Scale bar: 1 μm. Bottom 2: Zoomed-in HAADF-STEM and EDX images of the EOT cathode catalyst layer. Scale bar: 100 nm. **c** XRD patterns of the BOT and EOT cathode samples. **d** In-situ X-ray absorption spectra of the $Bi_2O_3$ electrode in 0.1 M KOH as a function of potential (0.8 V to −1.5 V vs RHE).

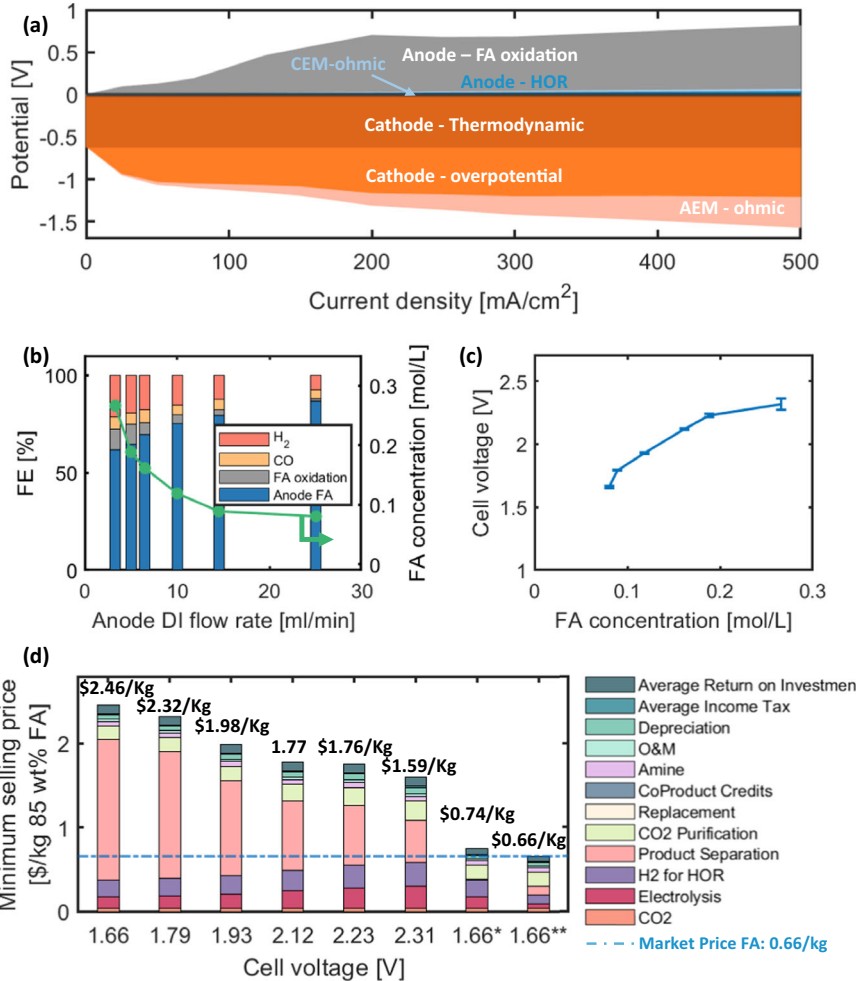

**Fig. 6 | Voltage break-down, operating parameters and their impacts. a** Cell voltage break-down using $H_2$ reference electrode of the cell operated at 60 °C with Pt/C anode and 80 μm AEM. **b** FE and formic acid concentrations collected at 200 mA/cm² with different anode DI water flow rate. **c** Cell voltage at 200 mA/cm² when different concentrations of formic acid are collected at the anode. Error bars stand for S.D. from three different measurements. **d** Minimum selling price break down based on the performance at different DI water flow rates, using industrial national average electricity price of 0.068 $/kWh, and 4.5 $/kg hydrogen. (*: Assuming the minimum amount of formic acid oxidation at the anode with 10 M FA concentration, industrial national average electricity price of 0.068 $/kWh, and 4.5 $/kg hydrogen. **: Assuming minimum amount of formic acid oxidation at the anode with 1.3 M FA concentration, projected future electricity price of 0.03 $/kWh, and 2.3 $/kg hydrogen. Dashed line represents market price for 85 wt% FA).

## Operating conditions and TEA analysis

To quantify exactly how much opportunity exists to improve the energy efficiency as formic acid oxidation is suppressed, a $H_2$ reference electrode was used to discern voltage loss contributions[39]. At current densities below 500 mA/cm², the cathode potential remained lower than −1.25 V. The anode potential is divided into two main parts, the theoretical overpotential of HOR predicted by the HOR exchange current density and Bulter-Volmer equation from previous measurements[40], and the remainder attributed to formic acid oxidation. A small rate of formic acid oxidation reaction at the anode can lead to a significant increase in the anode potential due to its much slower reaction kinetics compared to HOR[41]. The result indicates that nearly 500 mV of overpotential can be eliminated by completely supressing anode formic acid oxidation.

To test this assessment, the deionized water (DI) flow rate was varied on the anode inlet to reduce formic acid effluent concentration. Figure 6b and c show the resulting FE, formic acid concentrations and cell voltages at 200 mA/cm² as a function of anode DI flow rate. As the DI flow rate increased from 3.3 mL/min to 25 mL/min, formic acid anode concentration decreases from 0.27 to 0.08 mol/L. In comparison, when using a interlayer configuration as presented by Xia et al.'s work[30], the formic acid concentration obtained is 1.8 mol/L at 200 mA/cm². The reduced concentration improved overall formic acid FE, decreasing $H_2$ FE, as cathode pH becomes more basic due to reduced back diffusion of formic acid. The reduced formic acid concentration at the highest DI flow rate also nearly eliminates formic acid oxidation, pushing the overall cell voltage to just below 1.7 V at 200 mA/cm². Cell temperature can also affect the overall performance, with results shown in Fig. S10. Nevertheless, whether through the use of anode catalysts with improved $H_2$ to formic acid selectivity, or device operation, PCEM-based architectures achieve vastly improved energy efficiency when formic acid oxidation is supressed.

A techno-economic analysis (TEA) was conducted to obtain the minimum selling price for FA across a range of operating conditions, shown in Fig. 5d. The approach and inputs of the TEA can be found in SI. When FA concentration is higher at the anode exhaust, the overall cost for FA is lower due to reduced separation cost, despite a higher cell voltage. If anode formic acid oxidation can be minimized through catalyst development or electrode engineering, the combination of lower cell voltages (1.66 V) and higher effluent FA concentration (10 M) would enable electrochemical FA production cost as low as 0.74 $/kg with electricity price of 0.068 $/kWh and 4.5 $/kg hydrogen[42].

Furthermore, when coupled with renewable electricity with projected future cost of 0.03 \$/kWh, and hydrogen cost of 2.3 \$/kg, FA effluent targets would be relaxed to 1.3 M with a final projected production cost of 0.66 \$/kg[43]. This is comparable with the current market price. Consequently, future efforts with focus on materials and electrode structures can further reduce the anode oxidation, while simultaneously enabling operation at lower cell voltage to generate higher concentration of FA.

In summary, we investigated several zero-gap MEA configurations for $CO_2$ to formic acid reduction and proposed a structure that contains a composite forward bias bipolar membrane including a perforated cation exchange membrane (PECM) to facilitate the mass transport of formic acid generated at the membrane interface. This configuration generated >96% formic acid up to 0.25 M (using an anode DI flow rate of 3.3 mL/min). At higher DI flow rates (25 mL/min), the configuration yielded >80% FE 200 mA/cm$^2$ current density at 1.7 V using a 25 cm$^2$ cell. At moderated anode DI flow rates (10 mL/min) the PECM configuration maintained a stable voltage and high FE for formic acid over a 55 hr test at 200 mA/cm$^2$. The high stability and selectivity achieved with commercially available catalysts and polymer membrane materials can be further amplified by combining it with optimized electrocatalysts. Subsequent efforts will focus on tuning operating conditions, anode catalyst selectivity, and MEA structure to reduce formic acid oxidation, enabling a more concentrated effluent stream at lower cell voltages. The simplistic approach for $CO_2$ for formic acid presented here, eliminates the need for anolyte and catholyte chambers, interlayer components, and specialized materials thereby improving cell energy efficiency and reducing system complexity to facilitate system scale-up. This proposed configuration provides a platform for future development of technically and economically viable $CO_2$ conversion devices.

## Methods

### Cathode gas diffusion electrode fabrication

All materials and reagent grade solvents were used as received unless otherwise noted. Bismuth oxide catalyst ($Bi_2O_3$, 80 nm) was purchased from US Research Nanomaterials, Inc. The polymer (AP1-CNN8-00-X) powder was supplied by IONOMR. Omnisolv® grade *n*-propanol (*n*PA) and ultra-pure water (18.2 Ω, Milli-Q® Advantage A10 Water Purification System) were obtained from Millipore Sigma. ACS Certified methanol and acetone were purchased by VWR Chemicals BDH® and Fisher Chemical, respectively. The polymer powder was combined in a 1:1 weight mixture of acetone and methanol to render a 6.5 wt.% polymer dispersion. The catalyst ink was prepared by combining 20 g of $Bi_2O_3$, ultra-pure water, *n*PA, and ionomer dispersion into a 30 mL jar. This formulated recipe contained 30 wt.% catalyst, an ionomer-to-catalyst weight ratio of 0.02, and an alcohol-to-water weight ratio of 2:3 (40 wt.% *n*PA). 70 g of Glen Mills 5 mm zirconium oxide grinding media were added to this mixture prior to mixing. Samples were placed on a Fisherbrand™ Digital Bottle Roller at 80 rpm for 26 hrs. Inks were allowed to settle for 20 min before coating. $Bi_2O_3$ inks were coated at 22 °C utilizing a ½" x 16" wire wound lab rod (RD Specialties – 60 mil diameter) on a Qualtech automatic film applicator (QPI-AFA6800). 5 mL of catalyst ink deposited on 7.5" x 8" Sigracet 39 BB carbon gas diffusion media (Fuel Cell Store) were rod coated at a fixed average speed of 55 mm/sec. These coated electrodes were transferred to an oven and dried at 80 °C. The rod coating process and the image of coated GDE are shown in Fig. S4a and b. The coated GDE has a loading of 3.0 mg, $_{Bi_2O_3}$/cm$^2$, as confirmed by X-ray fluorescence (XRF) instrument (Fischerscope® XDV-SDD, Fischer-Technolgy Inc. USA).

### Membrane electrode assembly

For the composite membrane configuration that contains anion exchange membrane (AEM) and perforated CEM. Nafion NC700 (Chemours, USA) with a nominal thickness of 15 μm is used as the CEM

layer. The anode catalyst was directly spray-coated on the CEM, with an ionomer-to-carbon ratio of 0.83, and a coating area of 25 cm$^2$. High surface area supported platinum (50 wt.% Pt/C, TEC 10E50E, TANAKA precious metals) with a loading of 0.25 mg,$_{Pt}$/cm$^2$ was used as the anode catalyst. Nafion D2020 (Ion Power, USA) was used as the anode catalyst layer ionomer. The CEM perforation was conducted by cutting parallel lines on the CEM membrane with a spacing of 3 mm. Details of the perforation process is illustrated in Fig. S12b and c. The perforation has a gap of 32.6 μm, confirmed by X-ray computed tomography, as shown in Fig. S12d and e. During cell assembly, the perforated catalyst-coated CEM membrane was place on a 25 cm$^2$ Toray paper (5 wt.% PTFE treated, Fuel Cell Store, USA). An AEM membrane (PiperION, Versogen, USA) with thickness of either 25, 40, or 80 μm was placed on top of the CEM and then the cathode GDE. The AEM membrane is cut to a size of 7.5×7.5 cm to cover the entire flow field and soaked overnight in 1 M potassium hydroxide solution before assembly. A PTFE gasket was used for both anode and cathode with thicknesses to achieve an optimum GDE compression of 18%. Details of the cell assembly process is shown in Fig. S12a.

**Electrochemical Testing.** During the test, the assembled cell was held at 60 °C (for temperature-dependent study, 30, 60 and 80 °C), with the anode supplied with 0.8 slpm hydrogen and cathode supplied with 2 slpm $CO_2$ gas. Both anode and cathode gas stream were humidified with a relative humidity of 100%, and a cathode absolute pressure of 259 kPa. During the operation, the cathode gas stream was mixed with 2 mL/min of 1 molar KOH, to facilitate cathode catalyst layer utilization and ion conduction. The anode gas stream was mixed with 10 mL/min DI water, to help remove formic acid at the anode. Detailed information about the cell inputs and outputs are shown in Figure S5. The cathode effluent gas contains $CO_2$ and generated CO and $H_2$, and it will pass through a condenser (low-temperature heat exchanger with a temperature of 2 °C) to get rid of water vapor. The remainder of the gas will be collected for gas chronography analysis. The anode effluent will also go through the condenser to separate the liquid from the gas. The liquid effluent will be collected in a clean vial and then get analyzed using liquid chronography to quantify the formic acid generated. The electrochemical testing was conducted using a Garmy Potentiostat (Reference 30K, Gamry, USA). The cell was conditioned using liner scanning voltammetry from 0 to 250 mA/cm$^2$ for 4 times with a scan rate of 2.5 mA/cm$^2$ prior to polarization curve measurement. The polarization curves were obtained using galvanostatic mode, with the cell held at certain current densities for 4 mins before collecting cathode gas and anode liquid sample.

**Reference electrode and voltage breakdown.** We used a hydrogen reference electrode in the MEA to separate the cathode and anode potentials. The structure of the reference electrode is shown in figure S6a. A strip of Nafion membrane (Nafion 211, IonPower, USA) is used as the ion bridge to connect the MEA membrane with the reference electrode. One end of the Nafion strip was connected to a 1 cm$^2$ gas diffusion electrode (GDE) that has a loading of 0.25 mg Pt/cm$^2$ (50wt.% Pt/C, TEC10E50E, TANAKA precious metals) spray coated on 29BC carbon paper (Fuel Cell Store, USA). Custom-made Polyether ether ketone (PEEK) hardware was used for gas sealing and to guarantee good contact between the GDE and the Nafion strip, as well as to connect the reference electrode to the fuel cell hardware. The other end of the Nafion strip was linked to an overhanging edge of the cell's CEM. Figure S6b presents a cross-sectional view of the integration of the reference electrode with the MEA.

**Product quantification.** The gas samples were collected from the cathode, after the effluent gas passed through the condenser and gas-liquid separator. The collected gas was analyzed using a 4900 Micro GC (10 m, molecular sieve, Agilent) at least three times. Samples were

collected in Supel™ Inert Multi-Layer Foil Gas Sampling Bags (Sigma-Aldrich) for a specific duration (30 seconds) and manually inserted into the Micro GC within two hours of collection. The injection temperature was set to 110 °C. Carbon monoxide (CO) and hydrogen ($H_2$) were separated within a heated (105 °C) and pressurized (28 psi) 10 m MS5A column using argon (Matheson Gas- Matheson Purity) as the carrier gas. The compounds were detected on an integrated thermal conductivity detector (TCD). The GC chromatogram and calibration curves of CO and $H_2$ are shown in Figure S7. Liquid formic acid sample were collected from the anode with a certain period (120 seconds) and filtered with PTFE 0.22 μm syringe filters into 2 mL vials. The liquid products in the vials were analyzed using an Agilent 1260 Infinity II Bio-inert High-Performance Liquid Chromatography (HPLC) system, where a 20 μL sample volume was injected via autosampler (G5668A) with a mobile phase of 4 mM sulfuric acid ($H_2SO_4$) flowing at 0.6 mL/min (G5654A quaternary pump). The products were separated on a heated (35 °C, G7116A column thermostat) Aminex HPX-87H 300 × 7.8 mm Column (Bio-Rad) with a preceding Micro-Guard Cation H guard column. Formic acid was detected on a Diode Array Detector (DAD) at 210 nm with a bandwidth of 4 nm. The HPL chromatogram and the calibration curve of the formic acid standards are presented in figure S7.

**Faradaic efficiency, formate/FA yield, and formic acid purity calculation.** Liquid product (Formic acid) FE is calculated using Eq. 1,

$$FE_{formic\,aicd} = \frac{n_i * F * C_i * V}{j * A * t} \quad (1)$$

The gas product (CO and $H_2$) FE are calculated using the following equation, with the total mol of gas calculated using ideal gas equation:

$$FE_{gas} = \frac{n_i * F}{j * A * t} * \frac{P * x_i}{R * T} \quad (2)$$

Where: $n_i$: number of electrons for the electrochemical reaction. $F$: Faraday's constant. $C_i$: Concentration of the liquid product from HPLC. $V$: Volume of the collected liquid sample over fixed time $t$. $j$: Current density. $A$: Geometric area of the electrode (25 cm$^2$). $t$: Time period for the sampling. $P$: Absolute pressure. $x_i$: gas mol% as quantified by the GC. $R$: gas constant. $T$: Temperature.

The anode cation concentrations are quantified using inductively coupled plasma atomic emission spectroscopy (ICP-OES). Cations that could leach or diffuse into the anode include, Ti, Pt, Bi and K. All the other cations were below the detection limit except K. The formate ion in the anode effluent solution is either paired with proton or other cations. Thus, the formic acid purity can be calculated as

$$P_{FA} = \frac{C_{HCOO-} - C_{K+}}{C_{HCOO-}} \times 100\% \quad (3)$$

Formate/FA yield is the amount of FA generated per kWh of electricity consumed using a certain MEA configuration, with unit mol/kWh. It is calculated based on the current density, cell voltage, and Faradaic Efficiency at certain operating conditions.

$$Yeild_{formic\,aicd} = \frac{I}{nF} \times \frac{FE_{formic\,acid}}{I * V} \quad (4)$$

The amount of formic acid oxidation at the anode is calculated based on the total mass balance. There are three competing reactions at the cathode, hydrogen evolution reaction, $CO_2$ reduction to CO and $CO_2$ redcution to formic acid. Because we have formic acid oxidation process at the andoe, the FE for formic acid can be further divided into two parts, formic acid collected and formic acid oxidized. The total

mass balance can be written as below:

$$FE_{FA,collected} + FE_{FA,oxidized} + FE_{HER} + FE_{CO} = 100\% \quad (5)$$

We quantified the amount of formic acid collected from HPLC, the amount of hydrogen and CO using GC. It should be noted that the majority of the formic acid is collected from the anode, using setup depicted in supplemental figure S5. The is the negligible amount of formate collected from the cathode compartment, about two orders of magnitude smaller and accounts for less than 0.5% of the total FE.

**Model description.** The continuum transport model used here is based on a previous work for a similar system[34]. The coupled Poisson-Nernst-Planck (PNP) system of equations were used to solve for the aqueous species concentrations, as well as the electrostatic potential in both electronic and ionic conducting phases. A detailed overview of the governing equations and model geometry is given in the SI.

The system is solved for the concentrations of eight aqueous species ($CO_{2(aq)}$, $H^+$, $OH^-$, $HCO_3^-$, $CO_3^{2-}$, $HCOOH$, $HCOO^-$, and $K^+$), the electrostatic potential in the ion conducting phases ($\phi_I$), and the electrostatic potentials in the anode and cathode electronic conducting phases ($\phi_A$ and $\phi_C$, respectively). Neither local electroneutrality nor charge distribution functions were implemented; rather the space charge regions were resolved directly using the Poisson equation. This approach allowed us to directly model the Donnan exclusion effect at the CEM|AEM, CEM|Pore, and AEM|Pore interfaces. In addition, porous electrode theory (PET) was used to describe the charge transfer in the anode and cathode catalyst layer. To the authors' knowledge, this work represents the first application of PET to systems with several space charge regions.

**Nano-X-ray computed tomography (nano-CT).** BOT and EOT cathode GDE samples were tested using Zeiss Xradia 800 Ultra, with an X-ray source of 8.0 keV, absorption and large field of view mode with image binning 1. 901 images were collected from angle −90° to 90° and exposure time of 50 s. The reconstruction was performed using the filter back projection method and has a voxel size of 64 nm. The segmentation and particle size distribution analysis were conducted with custom-written code.

**HAADF STEM.** The electron microscopy characterizations involved embedding tested MEAs in epoxy in preparation for diamond knife ultramicrotomy. Cross sections of each MEA were cut to a thickness between 50 and 75 nm. The Talos F200X transmission electron microscope (Thermo Fisher Scientific) was used for scanning transmission electron microscopy (STEM) and energy-dispersive X-ray spectroscopy (EDS) measurements. The microscope was equipped with Super-X EDS system with 4 SDD windowless detectors and operated at 200 kV.

**XRD characterizations.** Powder X-ray diffraction (PXRD) patterns were acquired on a Bruker Advance D8 Powder X-ray Diffractometer with Ni-filtered Cu Kα radiation operating at 40 kV and 40 mA. The scan range was between 10° and 60° with 0.005° steps at a collection speed of 1 s/step.

**In-situ aqueous cell.** A home-made cell was utilized for measuring XAS spectra at the Bi $L_3$ edge for the $Bi_2O_3$ catalyst as a function of potential. A $Bi_2O_3$ catalyst-ionomer ink was prepared using 26.1 mg $Bi_2O_3$ mixed with 156.3 uL of ionomer solution (6.68%), neutralized with 1 M KOH, water (157 uL), and isopropanol (104 uL) giving an ionomer-to-catalyst ratio of 0.4. The ink was deposited in rectangular spots (10 mm×4 mm) onto a graphene sheet to achieve a catalyst loading of 0.5 mg/cm$^2$ of $Bi_2O_3$. The remainder of the graphene sheet was covered with Kapton to insulate these areas from the electrolyte. The catalyst-coated

graphene sheet was inserted between two PTFE and screwed onto the cell body (PEEK), Figure S8. The reference electrode was Hg/HgO (1 M NaOH) and the counter electrode was carbon paper. The Hg/HgO reference electrode was calibrated versus a Pt wire immersed in hydrogen-saturated 0.1 M KOH for the conversion of all measured potentials to the reversible hydrogen electrode (RHE) scale. The XAS spectra were acquired while controlling the potential of the $Bi_2O_3$/graphene sheet working electrode immersed in 0.1 M KOH heated to 30 °C. The electrolyte was circulated through the cell, with electrolyte inlet at the bottom on the cell and outlet at the top to ensure electrolyte contact with the catalyst layer in the event of bubble formation. The potential of the working electrode was controlled using a CH Instruments 760e potentiostat. The potential sequence was open circuit potential, −100, −200, −300, −400, −500, −800, −850, −900, −1000, −1100, −1500, and +700 mV vs RHE. All potentials have been iR corrected.

**XAFS experiments.** Bi $L_3$ edge (~13424 eV for Bi metal) X-ray absorption fine structure (XAFS) spectra were measured in fluorescence mode at the Materials Research Collaborative Access Team (MRCAT) beam line 10-ID, Advanced Photon Source (APS), Argonne National Laboratory. The X-ray energy was tuned using a liquid nitrogen-cooled Si(111) double crystal monochromator, while the harmonic content was attenuated using a Rh-coated mirror. The scan energy range was 13200 eV to 14400 eV, and the fluorescence was measured using a 5 × 5 silicon PIN diode grid array without filters or Soller slits. The energy was calibrated via the $L_2$ edge for a Pt foil to 13271.90 eV for the zero-crossing of the second derivative. Due to the thickness of the electrochemical cell, the spectrum for a reference standard was not measured simultaneously. Therefore, the estimated scan-to-scan variation of the incident X-ray energy was ±0.015 eV, based on repeated measurements over the course of the experiment. The thickness of the $Bi_2O_3$ layer resulted in some fluorescence self-absorption; the electrode remained in a fixed orientation with respect to the incident beam and detector, rendering the effect nearly identical for all scans. The near-edge regions of the XAFS spectra were utilized to determine the oxidation state and chemical speciation of bismuth by the comparison with the XANES region of Bi and $Bi_2O_3$ standards using linear combination fitting algorithm of the Athena software (version 0.9.26), based on the IFEFFIT code[44].

## Data availability
The data that support the plots within the paper and other findings of this study are available from the corresponding authors upon reasonable request.

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

## Acknowledgements

This work was funded by the Bio Energy Technology Office, authored in part by Alliance for Sustainable Energy, LLC, the manager, and operator of the National Renewable Energy Laboratory for the U.S. Department of Energy (DOE) under Contract No. DE-AC36-08GO28308. The research was supported by the U.S. Department of Energy, Office of Energy Efficiency and Renewable Energy, Bioenergy Technologies Office, under contract DE-AC02-06CH11357 (Argonne National Laboratory). Neither the U.S. Government nor any agency thereof, nor any of their employees or employees of contributing companies, makes any warranty, expressed or implied, or assumes any legal liability or responsibility for the accuracy, completeness, or usefulness of any information, apparatus, product, or process disclosed, or represents that its use would not infringe privately owned rights. This work was authored in part by Argonne National Laboratory, a DOE, Office of Science Laboratory operated under Contract No. DE-AC02-06CH11357 by UChicago, Argonne, LLC. The X-ray spectroscopy experiments were performed at beamline 10-ID at the APS, which is operated by the Materials Research Collaborative Access Team (MRCAT). MRCAT operations are supported by the Department of Energy and the MRCAT member institutions. The APS is a U.S. Department of Energy (DOE) Office of Science User Facility operated for the DOE Office of Science by Argonne National Laboratory under Contract No. DE-AC02-06CH11357. The authors would like to thank Joshua Wright of 10-ID. The views expressed in the article do not necessarily represent the views of the DOE or the U.S. Government. The U.S. Government retains and the publisher, by accepting the article for publication, acknowledges that the U.S. Government retains a non-exclusive, paid-up, irrevocable, worldwide license to publish or reproduce the published form of this work, or allow others to do so, for U.S. Government purposes.

## Author contributions

L.H.: Conceptualization; Experiment; Formal Analysis; Investigation; Methodology; Validation; Visualization; Writing – original draft. J.A.W.: Software; Resources; Investigation; Writing – review & editing. C.B.-C.: Investigation; Writing – review & editing. F.I.: Methodology; Formal Analysis; Investigation; Writing – review & editing. J.H.P.: Formal Analysis; Resources; Investigation; Writing – review & editing. Author J.K.: Resources; Investigation; Writing – review & editing. N.K.: Formal Analysis; Investigation; Writing – review & editing. Z.H.: Investigation; Writing – review & editing. A.F.: Resources; Investigation; Writing – review & editing. L.A.: Formal Analysis; Resources; Investigation; Writing – review & editing. PS.: Investigation; Writing – review & editing. LT.: Investigation; Writing – review & editing. D.A.C.: Supervision; Investigation; Writing – review & editing. D.J.M.: Supervision; Visualization; Investigation; Writing – review & editing. M.S.F.: Formal Analysis; Supervision; Visualization; Investigation; Writing – review & editing. K.C.N.: Conceptualization; Project administration; Funding acquisition; Investigation; Writing – review & editing; Supervision; Project administration.

## Competing interests

The authors declare no competing interests.
