## [Peer Review File · Nature Communications]

REVIEWER COMMENTS

Reviewer #1 (Remarks to the Author):

The manuscript titled “A novel, scalable membrane electrode assembly architecture for efficient electrochemical conversion of CO₂ to formic acid” presents a significant advancement in the field by introducing an innovative architecture that enables the direct synthesis of formic acid from carbon dioxide through the perforation of a cationic exchange membrane. The obtained results are highly promising and intriguing, demonstrating the ability to achieve formic acid concentrations of up to 0.25 M. Additionally, the manuscript includes stability tests conducted over a 55-hour period, which exhibit consistent Faradaic Efficiencies values and stable cell voltage. Moreover, the authors have taken a commendable step by performing a techno-economic analysis to provide insights into the potential cost competitiveness of their approach compared to existing formic acid production methods. This analysis enhances the manuscript's value by illustrating a viable pathway towards achieving cost parity in formic acid production. The novelty of the proposed architecture is evident, and the manuscript holds significant potential for publication in Nature Communications. However, to ensure the manuscript's quality, it is essential to address the following comments and suggestions:

- 1) I would recommend including an explanation of the state-of-the-art in the field of CO₂ electroreduction to obtain formic acid or formate (as well the Table S1 of the Supporting Information), supported by important references. Here are some suggested references:
 - i) Diaz-Sainz et al., Improving trade-offs in the figures of merit of gas-phase single-pass continuous CO₂ electrocatalytic reduction to formate, *Chemical Engineering Journal*, 405, 2021, 126965.
 - ii) Proietto et al, Electrochemical conversion of CO₂ to HCOOH at tin cathode in a pressurized undivided filter-press cell, *Electrochimica Acta*, 277, 2018, 30-40.
 - (iii) Tran-Phu et al., Nanostructured β -Bi₂O₃ Fractals on Carbon Fibers for Highly Selective CO₂ Electroreduction to Formate, *Advanced Functional Materials*, 30, 2020, 1-8.
- 2) This aspect is of great significance and calls for further elaboration. The author should conduct a comprehensive analysis of how their work enhances the scalability of the CO₂ electroreduction process. It is crucial to provide insights into how the proposed method can be effectively scaled up to industrial levels and discuss the potential advantages or challenges associated with such scaling.
- 3) Further elaboration on Figure 1 is warranted. The authors should place greater emphasis on the main reactions occurring in the cathodic and anodic compartments within the different configurations depicted in Figure 1.
- 4) It is recommended to include and discuss a recent review article in the revised version of the manuscript to provide a comprehensive overview of the current state of research in the field. One such article that merits consideration is "Electroreduction of CO₂: Advances in the Continuous Production of Formic Acid and Formate" by Fernández-Caso et al. (*ACS Energy Letters*, 8(4), 2023, 1992-2024).

- 5) It is crucial to provide clarity on how the authors ensure a relative humidity of 100% in the cathode gas stream. Controlling the amount of water in the CO₂ input stream is one potential approach to achieve this, but further details are required. To enhance the understanding of the experimental setup and procedures, it is recommended to include a schematic diagram of the setup in the manuscript.
- 6) It is essential to discuss how the authors estimate the oxidation of formic acid, considering that it is collected in both compartments of the MEA structure with an anion exchange membrane (AEM) and a perforated cation exchange membrane (CEM). Providing insights into the methodology used to estimate the oxidation of formic acid will enhance the understanding of the experimental process and the reliability of the results.
- 7) The authors are suggested to provide the cathode and anode potential in order to compare with others' work.
- 8) The calculus of figures of merit (e.g. rate, energy efficiency) should be included in the methodology section.
- 9) Can you explain the advantages of using an anion exchange membrane (at the beginning of the work) instead of a cation exchange membrane?
- 10) The loading of the catalyst in both the anode and cathode is an important parameter that can significantly impact the performance of the CO₂ electroreduction process. The catalyst loading refers to the amount of catalyst material deposited on the electrode surface. It plays a crucial role in determining the catalytic activity, selectivity, and overall efficiency of the electrochemical reactions.
- 11) The authors are encouraged to clearly identify the uniqueness of their manuscript in comparison to other works and adjust their claims accordingly.
- 12) The structure of the results section in the study could be improved with more sub-sections to enhance readability. Additionally, providing a more comprehensive explanation of the different phenomena occurring throughout the process would help readers better understand the results.
- 13) While the manuscript focuses on the energy consumption aspect of the CO₂ electroreduction process, it is important to also address the formic acid/formate concentration achieved in comparison to the existing literature. The concentration of formic acid or formate is a critical parameter as it directly relates to the efficiency and practicality of the process for potential applications. Therefore, it is recommended to include a discussion regarding the formic acid/formate concentrations obtained in this study and their comparison to those reported in the literature. By highlighting the concentration values achieved in the current work and comparing them to the reported concentrations in previous studies, the authors can provide insights into the performance and competitiveness of their proposed method.
- 14) It is recommended to provide additional details about the perforation process of the cation exchange membrane in order to enhance the readers' understanding. It would be beneficial to include illustrative figures or diagrams that depict the steps involved in the perforation process.

Reviewer #2 (Remarks to the Author):

In this work, the authors developed a zero-gap membrane electrode assembly (MEA) cell configuration using a perforated cation exchange membrane (PCEM) for the conversion of CO₂ to formic acid. By incorporating the PCEM layer, they successfully addressed the issues associated with the forward bias operation of a bipolar membrane for CO₂ reduction. The authors utilized Bi-based catalysts for CO₂ reduction and Pt/C catalysts for hydrogen oxidation at the anode, resulting in a formic Faradaic efficiency of over 80% at a current density of 300 mA/cm². Their system produces formic acid, which is collected at the anode side, and remains stable for 55 hours at a current density of 200 mA/cm². When H₂ oxidation is employed at the anode, the cell voltage decreases to below 2 V at a current density of 200-300 mA/cm². Overall, this work presents an interesting approach to membrane modification for CO₂ reduction to formic acid and potentially has a significant impact. I have some comments and questions below:

1. Details about the "pore" structure of the PCEM layer should be provided. How do the pore size and density affect the diffusion of formic products and the overall performance?
2. The quantification and discussion of CO₂ crossover through the system should be included. It is unclear if CO₂ crossover has been included in the TEA model.
3. It is interesting that water and formic acid do not cause flooding in the carbon layer of the anode. Why did the author use carbon paper with 5% PTFE content?
4. Could the authors explain why the perforation was employed only on CEM and not on both the carbon paper and CEM? The latter configuration may enhance the diffusion of formic acid from the CEM layer and prevent further oxidation.

Reviewer #3 (Remarks to the Author):

In this manuscript, a novel CO₂RR electrolyzer was designed by introduced a perforated cation exchange membrane (PECM) to facilitate the mass transport of formic acid generated at the membrane interface. The PECM cell exhibited good stability and high formic acid selectivity under high current. Authors also demonstrated the value of this technique through a techno-economic analysis. Though this novel structure of CO₂RR electrolyzer is of interest for the electrocatalytic community, the reviewer thinks the study and the presented data are not comprehensive. Some questions and concerns need to be addressed before the manuscript can be assessed for publication.

1. The control device using a cation exchange membrane should be constructed. And the issue of this kind of device also should be discussed.
2. Detailed information about the PCEM perforated cation exchange membrane, such as optical pictures and its preparation method, should be provided. The author should also provide physical

pictures of each component of the equipment, and mark the function of each component in the actual working equipment.

3. Authors should conduct a systematic investigation on the pore size and distribution of cation exchange membrane on the performance of PCEM.

4. Figure S3 shows the durability of the system with an 80 mm AEM and perforated CEM at 60°C, What is the reason for the periodic fluctuation of the data in the figure?

5. How is formic acid effectively collected in actual devices .

6. Is it possible to quantify the proportion of formic acid oxidation current in the total anodic current.

Manuscript ID: NCOMMS-23-22685-T

Title: A Novel, Scalable Membrane Electrode Assembly Architecture for Efficient Electrochemical Conversion of CO₂ to Formic Acid

Response to reviewer comments

We would like to extend our thanks to the reviewers for putting their valuable time into this review. Their comments and suggestions have been valuable in improving the manuscript. We have made changes in response to the reviewers' comments. Our **responses** are in red and **text changes** are highlighted.

Reviewer #1 (Remarks to the Author):

The manuscript titled “A novel, scalable membrane electrode assembly architecture for efficient electrochemical conversion of CO₂ to formic acid” presents a significant advancement in the field by introducing an innovative architecture that enables the direct synthesis of formic acid from carbon dioxide through the perforation of a cationic exchange membrane. The obtained results are highly promising and intriguing, demonstrating the ability to achieve formic acid concentrations of up to 0.25 M. Additionally, the manuscript includes stability tests conducted over a 55-hour period, which exhibit consistent Faradaic Efficiencies values and stable cell voltage. Moreover, the authors have taken a commendable step by performing a techno-economic analysis to provide insights into the potential cost competitiveness of their approach compared to existing formic acid production methods. This analysis enhances the manuscript's value by illustrating a viable pathway towards achieving cost parity in formic acid production. The novelty of the proposed architecture is evident, and the manuscript holds significant potential for publication in Nature Communications. However, to ensure the manuscript's quality, it is essential to address the following comments and suggestions:

Comment 1. I would recommend including an explanation of the state-of-the-art in the field of CO₂ electroreduction to obtain formic acid or formate (as well the Table S1 of the Supporting Information), supported by important references. Here are some suggested references:

- i) Diaz-Sainz et al., Improving trade-offs in the figures of merit of gas-phase single-pass continuous CO₂ electrocatalytic reduction to formate, Chemical Engineering Journal, 405, 2021, 126965.
- ii) Proietto et al, Electrochemical conversion of CO₂ to HCOOH at tin cathode in a pressurized undivided filter-press cell, Electrochimica Acta, 277, 2018, 30-40.
- (iii) Tran-Phu et al., Nanostructured β -Bi₂O₃ Fractals on Carbon Fibers for Highly Selective CO₂ Electroreduction to Formate, Advanced Functional Materials, 30, 2020, 1-8.

Response: Thanks for your suggestions. We've updated our introduction section and included more state-of-the-art CO₂ electroreduction devices. For recommended references, only reference (i) contains the energy consumption info, ref (ii) and ref (iii) either doesn't have energy consumption numbers or doesn't contain info about cell voltage for energy consumption calculations. We also updated our Table S1 and the Figure 1b based on ref (i) data.

Changes:
(Line 37)

With momentum building for a formic acid economy^{1,9}, multiple research efforts have focused on the optimization of catalyst selectivity¹⁰⁻¹⁶. However, the majority of efforts remain on small-scale H-cell or liquid flow cells operating at low current densities (<50 mA/cm²). To reduce cost

(Line 48)

diffusion electrode (GDE) based CO₂R to formate/formic acid devices. These device configurations can be categorized into three main groups: 1. Flowing catholyte¹⁸⁻²⁵, 2. Single membrane (either cation exchange membrane (CEM)²⁶ or anion exchange membrane (AEM))²⁷, and 3. Interlayer configurations²⁸⁻³⁰. Simplified cross sections for these configurations are shown in Figure 2a. For catholyte configurations, an electrolyte chamber is created between the membrane and the cathode GDE. The flowing catholyte serves to provide ionic access to the

(Line 58, 59)

mA/cm²²⁷. Díaz-Sainz *et al*²⁶ used a filter press setup with a single CEM membrane and can achieve 89% FE at current density of 45 mA/cm². Both approaches yielded formate as opposed

(Table S1)

	4		1.33	222.0	72.98	
Single CEM configuration	Cell area (cm ²)	Catalyst		Current density	Energy consumption (kWh/kmol)	η
	10	Bi		45	207.4	
	10			100	409.8	
	10			200	546.0	

(Figure 1b)

Figure 1. (a) Comparison of the three most prominent device configurations for CO₂R to formate/formic acid, along with the architecture proposed in this study. (b) Comparison of total current and formate/formic acid yield for catholyte configuration, interlayer configuration, single CEM configuration from literature as shown in supplemental Table S1 and our work. Hollow markers represent the production of formate salt solution, while solid markers indicate the production of formic acid. * Stands for configurations using hydrogen at the anode. (c) The structure of zero-gap MEA configuration using composite bipolar membrane with perforated cation exchange layer operating in forward-bias mode.

Comment 2. This aspect is of great significance and calls for further elaboration. The author should conduct a comprehensive analysis of how their work enhances the scalability of the CO₂ electroreduction process. It is crucial to provide insights into how the proposed method can be effectively scaled up to industrial levels and discuss the potential advantages or challenges associated with such scaling.

Response: Thanks for your comments. We've added more discussions about scalability of each configuration.

Changes:

(Line 121)

architectures used for fuel cell and water electrolyzer stacks, enabling a more rapid transition to scale. Catholyte configuration contains a catholyte flow chamber, which can lead to pressure imbalance between the gas and liquid phase especially at larger cell configurations. For interlayer configuration with porous liquid flow layer, significant efforts are needed to optimize the porous interlayer to prevent significant pressure drop, carbon dioxide build-up within the interlayer and cell failure when scaling up to larger cells. It is also very hard to fabricate stand-alone thin, porous interlayer at a large scale. In contrast, the proposed new configuration is a zero-gap MEA configuration and doesn't contain any liquid flow chamber or interlayer. The uniqueness of the proposed configuration compared to other existing electrochemical cells is that it can generate formic acid instead of formate in a zero-gap configuration, and the structure is energy efficient and easy to scale-up.

Comment 3. Further elaboration on Figure 1 is warranted. The authors should place greater emphasis on the main reactions occurring in the cathodic and anodic compartments within the different configurations depicted in Figure 1.

Response: Thanks for your suggestions, we've added the corresponding reactions in the cathodic and anodic compartments to Figure 1.

Changes:

(Figure 1)

Figure 2. (a) Comparison of the three most prominent device configurations for CO₂R to formate/formic acid, along with the architecture proposed in this study. (b) Comparison of total current and formate/formic acid yield for catholyte configuration, interlayer configuration, single CEM configuration from literature as shown in supplemental Table S1 and our work. Hollow markers represent the production of formate salt solution, while solid markers indicate the production of formic acid. * Stands for configurations using hydrogen at the anode. (c) The structure of zero-gap MEA configuration using composite bipolar membrane with perforated cation exchange layer operating in forward-bias mode.

Comment 4. It is recommended to include and discuss a recent review article in the revised version of the manuscript to provide a comprehensive overview of the current state of research in the field. One such

article that merits consideration is "Electroreduction of CO₂: Advances in the Continuous Production of Formic Acid and Formate" by Fernández-Caso et al. (ACS Energy Letters, 8(4), 2023, 1992-2024).

Response: Thank you for your suggestions. We've added more discussions related to current state of research in this field.

Changes:

(Line 46-49)

Along these lines, recent efforts have been made to develop industrially relevant gas diffusion electrode (GDE) based CO₂R to formate/formic acid devices. Fernández-Caso et al.²⁰ made a comprehensive review summarizing all the electrochemical cell configurations for continuous reduction of CO₂ to formic acid/formate. In general all existing configurations can be categorized into three main groups: 1. Flowing catholyte^{19,21-27}, 2. Single membrane (either

Comment 5. It is crucial to provide clarity on how the authors ensure a relative humidity of 100% in the cathode gas stream. Controlling the amount of water in the CO₂ input stream is one potential approach to achieve this, but further details are required. To enhance the understanding of the experimental setup and procedures, it is recommended to include a schematic diagram of the setup in the manuscript.

Response: Thanks for the suggestions. We've updated supplemental Figure S5 to include more details of our experimental setup. Basically, the anode and cathode gas went through two separate humidifier bottles. When the cell is operated at 60C, the water temperature inside the humidifier bottle is also controlled to be 60C. The gas will bubble through the humidifier bottle and bring water vapor out to the cell. When the bottle temp is the same as the cell, the gas relative humidity is 100% RH. The RH for the electrochemical station is calibrated using a Vaisala relative humidity meter.

Changes:

(Supplemental Figure S5)

Figure S1. Detailed information about all the inputs and outputs of the cell during electrochemical cell testing. The anode is supplied with a mixture of 0.8 slpm H₂ at 100% RH and 10 mL/min DI water. The cathode is supplied with 2 slpm 100% CO₂ gas mixed with 2 mL/min 1 molar KOH. Formic acid is collected at the anode, and the CO and H₂ are collected from the cathode.

Comment 6. It is essential to discuss how the authors estimate the oxidation of formic acid, considering that it is collected in both compartments of the MEA structure with an anion exchange membrane (AEM) and a perforated cation exchange membrane (CEM). Providing insights into the methodology used to estimate the oxidation of formic acid will enhance the understanding of the experimental process and the reliability of the results.

Response: Thanks for pointing this out. We did characterize the liquid coming from both compartment, anode and cathode using an experiment set up shown in updated supplemental Figure S5. There is trace amount of formate that can be detected from the cathode dropout, about two orders of magnitude smaller compared to the anode, and accounts for < 0.5% of the total FE. This is probably because AEM ionomer and AEM membrane are used close to the cathode, making the formate ion preferably electromigrated from the cathode to the PCEM/AEM interface, instead of accumulating at the cathode catalyst layer. Because of that, we can estimate the oxidation of the formic acid based on the mass balance. We've added discussions about the formic acid oxidation estimation.

Changes:

(Line 126-127)

loss) increase (Figure S1d), illustrating the zero-sum tradeoff. The amount of formate oxidation is estimated based on total mass balance, with details in the method section. The performance of

(Line 422 - 432)

The amount of formic acid oxidation at the anode is calculated based on the total mass balance. There are three competing reactions at the cathode, hydrogen evolution reaction, CO₂ reduction to CO and CO₂ reduction to formic acid. Because we have formic acid oxidation process at the anode, the FE for formic acid can be further divided into two parts, formic acid collected and formic acid oxidized. The total mass balance can be written as below:

$$FE_{FA, collected} + FE_{FA, oxidized} + FE_{HER} + FE_{CO} = 100\%$$

We quantified the amount of formic acid collected from HPLC, the amount of hydrogen and CO using GC. It should be noted that the majority of the formic acid is collected from the anode, using setup depicted in supplemental figure S5. There is negligible amount of formate collected from the cathode compartment, about two orders of magnitude smaller and accounts for less than 0.5% of the total FE.

Comment 7. The authors are suggested to provide the cathode and anode potential in order to compare with others' work.

Response: Thanks for the comments and suggestions. Measuring anode and cathode potential in a zero-gap membrane electrode assembly is much harder than in a liquid cell. However, we did conduct a MEA polarization curve tests with an in-house designed hydrogen reference electrode attached.

Supplemental Figure S6 shows our H₂ reference electrode configuration.

(Figure S6)

Figure 6a shows the potential break-down analysis using this hydrogen reference electrode.

(Figure 6a)

Comment 8. The calculus of figures of merit (e.g. rate, energy efficiency) should be included in the methodology section.

Response: Thanks for pointing this out. We have the Faradaic Efficiency and Purity calculation descriptions in the method section. We added the Formate/Formic acid yield (mol/kWh) calculation to the same section, as highlighted.

Changes:

(Line 359)

Faradaic Efficiency, Formate/FA Yield, and Formic Acid Purity Calculation:

(Line 382-385)

Formate/FA yield is the amount of FA generated per kWh of electricity consumed using certain MEA configuration, with unit mol/kWh. It is calculated based on the current density,

cell voltage, and Faradaic Efficiency at certain operating conditions.

$$Yield_{formic\ acid} = \frac{I}{nF} \times \frac{FE_{formic\ acid}}{I * V}$$

Comment 9. Can you explain the advantages of using an anion exchange membrane (at the beginning of the work) instead of a cation exchange membrane?

Response: Thanks for your comments. We tested using single CEM MEA configuration, similar to the work by Lee W, Kim Y E, Youn M H, et al. *Angewandte Chemie International Edition*, 2018, 57(23): 6883-6887. Results are added to the Supplemental Figure S1 e-g. When single CEM is used, cations like potassium, sodium or proton will migrate from the anode to the cathode through the CEM, while it is hard for formate to transport from the cathode to the anode through the CEM. Because of that, salt like potassium formate will accumulate and then get collected from the cathode. The initial performance of this system looks good, but the potassium formate accumulation at the cathode can lead to severe durability issues, like the salt will precipitate and block the cathode catalyst layer. After two hours of operation, significant amount of salt accumulation can be observed, especially at high current densities (>100 mA/cm²). Photo images of the GDE and flow field after two hours of operation at 200 mA/cm² is also included in the Supplemental Figure S12.

Changes:

(Line 112-115)

anolyte concentration is reduced to 0.1 M KOH, both cell voltage and formate oxidation (formate loss) increase (Figure S1d), illustrating the zero-sum tradeoff. The performance of using a MEA configuration and single CEM membrane is also investigated, with results shown in Figure S1 f,g. The formate FE collected from the cathode is >60% at 200 mA/cm² at beginning of test, but suffers from fast degradation within two hours due to cathode salt accumulation (Figure S12).

(Supplemental Figure S1)

(Supplemental Figure S12)

Figure S 4. Images of the (a) back of the cathode GDE and (b) flow field of the single CEM MEA after 2 hours of operation under 200 mA/cm^2 . Significant amount of salt accumulation can be observed.

Comment 10. The loading of the catalyst in both the anode and cathode is an important parameter that can significantly impact the performance of the CO₂ electroreduction process. The catalyst loading refers to the amount of catalyst material deposited on the electrode surface. It plays a crucial role in determining the catalytic activity, selectivity, and overall efficiency of the electrochemical reactions.

Response: Thanks for pointing this out. We've added the loading information.

Changes:

(Line 303-305)

speed of 55 mm/sec. These coated electrodes were transferred to an oven and dried at 80 °C. The rod coating process and the image of coated GDE are shown in Figure S4 a and b. The coated GDE has a loading of $3.0 \text{ mg}_{\text{Bi}_2\text{O}_3/\text{cm}^2}$, as confirmed by X-ray fluorescence (XRF) instrument (Fischerscope® XDV-SDD, Fischer-Technology Inc. USA)

Membrane Electrode Assembly – For the composite membrane configuration that contains

(Line 309-311)

on the CEM, with an ionomer to carbon ratio of 0.83, and a coating area of 25 cm^2 . High surface area supported platinum (50 wt.% Pt/C, TEC 10E50E, TANAKA precious metals) with a loading of $0.25 \text{ mg}_{\text{Pt}/\text{cm}^2}$ was used as the anode catalyst. Nafion D2020 (Ion Power, USA) was used as

Comment 11. The authors are encouraged to clearly identify the uniqueness of their manuscript in comparison to other works and adjust their claims accordingly.

Response: Thank you for your suggestions. We've added more descriptions about the uniqueness of this work compared to others.

Changes

(Line 97-99)

up to larger cells. The uniqueness of the proposed configuration compared to other existing electrochemical cells is that it can generate formic acid instead of formate in a zero-gap configuration, and the structure is energy efficient and easy to scale-up.

Comment 12. The structure of the results section in the study could be improved with more sub-sections to enhance readability. Additionally, providing a more comprehensive explanation of the different phenomena occurring throughout the process would help readers better understand the results.

Responses: Thanks for your suggestions. We've added more subsections to the results and discussions section to make it more organized.

Changes:

(Line 115)

Results and Discussions

Screening of zero-gap MEA configurations

To suppress H₂ evolution, a plethora of CO₂ reduction efforts have utilized MEA configurations and AEM membranes, coupled with high molarity electrolytes (e.g. 1-10 M KOH)

(Line 148)

Performance of MEA with PCEM/AEM configuration

Based on the performance and failure mechanisms of the above-mentioned configurations, a new MEA architecture was proposed as shown in Figure 2c and detailed further in **Error! Reference source not found.**³⁸. Here, the PCEM layer provides pathways for formic acid and anions to migrate from the

(Line 183)

Stability of MEA with PCEM/AEM configuration

To test the stability of this PCEM based architecture, the cell was held at 200 mA/cm² for 55 hrs. The overall results are displayed in **Error! Reference source not found.** with results from the first 3 hrs highlighted in Figure S3. When a Pt/C anode catalyst was used, the cell voltage increased dramatically within

(Line 245)

Operating conditions and TEA analysis

To quantify exactly how much opportunity exists to improve the energy efficiency as formic acid oxidation is suppressed, a H₂ reference electrode was used to discern voltage loss contributions³⁹. At current densities below 500 mA/cm², the cathode potential remained lower

Comment 13. While the manuscript focuses on the energy consumption aspect of the CO₂ electroreduction process, it is important to also address the formic acid/formate concentration achieved in comparison to the existing literature. The concentration of formic acid or formate is a critical parameter as it directly relates to the efficiency and practicality of the process for potential applications. Therefore, it is recommended to include a discussion regarding the formic acid/formate concentrations obtained in this study and their comparison to those reported in the literature. By highlighting the concentration values achieved in the current work and comparing them to the reported concentrations in previous studies, the authors can provide insights into the performance and competitiveness of their proposed method.

Response: Thanks for pointing this out. We have the concentration info included in the manuscript, as shown in Figure 6c. We added a very brief comparison with other existing works.

Figure 3. (a) Cell voltage break-down using H₂ reference electrode of the cell operated at 60 °C with Pt/C anode and 80 μm AEM. (b) FE and formic acid concentrations collected at 200 mA/cm² with different anode DI water flow rate. (c) Cell voltage at 200 mA/cm² when different concentrations of formic acid are collected at the anode. (d) Minimum selling price break down based on the performance at different DI water flow rate, using industrial national average electricity price of 0.068 \$/kWh, and 4.5 \$/kg hydrogen. (*: Assuming minimum amount of formic acid oxidation at the anode with 10M FA concentration, industrial national average electricity price of 0.068 \$/kWh, and 4.5 \$/kg hydrogen. **: Assuming minimum amount of formic acid oxidation at the anode with 1.3M FA concentration, projected future electricity price of 0.03 \$/kWh, and 2.3 \$/kg hydrogen. Dashed line represents market price for 85wt% FA)

Changes:

(Line 260 - 262)

concentrations and cell voltages at 200 mA/cm² as a function of anode DI flow rate. As the DI flow rate increased from 3.3 mL/min to 25 mL/min, formic acid anode concentration decreases from 0.27 to 0.08 mol/L. In comparison, when using a interlayer configuration as presented by Xia *et al.*'s work³⁰, the formic acid concentration obtained is 1.8 mol/L at 200 mA/cm². The reduced concentration improved overall formic acid FE, decreasing H₂ FE, as cathode pH becomes more basic due to reduced back diffusion of formic acid. The reduced formic acid concentration at the highest DI flow rate also nearly eliminates formic acid oxidation, pushing

Comment 14. It is recommended to provide additional details about the perforation process of the cation exchange membrane in order to enhance the readers' understanding. It would be beneficial to include illustrative figures or diagrams that depict the steps involved in the perforation process.

Response: Thanks for your suggestions. We've added a supplemental Figure S13 with details of the perforation process. We also updated the method section to include more descriptions of the perforation steps.

Changes:

(Supplemental Figure S13)

(Line 342-350)

used as the anode catalyst layer ionomer. The CEM perforation was conducted by cutting parallel lines on the CEM membrane with a spacing of 3 mm. Details of the perforation process is illustrated in Figure S13 b and c. The perforation has a gap of 32.6 μm , confirmed by X-ray computed tomography, as shown in Figure S13 d and e. During cell assembly, the perforated catalyst coated CEM membrane was placed on a 25 cm² Toray paper (5 wt.% PTFE treated, Fuel Cell Store, USA). An AEM membrane (PiperION, Versogen, USA) was placed on top of the

CEM and the cathode GDE on top of the AEM. A PTFE gasket was used for both anode and cathode with thicknesses to achieve an optimum GDE compression of 18%. Details of the cell assembly process is shown in Figure S13 a.

Reviewer #2 (Remarks to the Author):

In this work, the authors developed a zero-gap membrane electrode assembly (MEA) cell configuration using a perforated cation exchange membrane (PCEM) for the conversion of CO₂ to formic acid. By incorporating the PCEM layer, they successfully addressed the issues associated with the forward bias operation of a bipolar membrane for CO₂ reduction. The authors utilized Bi-based catalysts for CO₂ reduction and Pt/C catalysts for hydrogen oxidation at the anode, resulting in a formic Faradaic efficiency of over 80% at a current density of 300 mA/cm². Their system produces formic acid, which is collected at the anode side, and remains stable for 55 hours at a current density of 200 mA/cm². When H₂ oxidation is employed at the anode, the cell voltage decreases to below 2 V at a current density of 200-300 mA/cm². Overall, this work presents an interesting approach to membrane modification for CO₂ reduction to formic acid and potentially has a significant impact. I have some comments and questions below:

1. Details about the "pore" structure of the PCEM layer should be provided. How do the pore size and density affect the diffusion of formic products and the overall performance?

Response: Thank you for your comments and suggestions. We've added characterizations to the pore structure in the supplemental Figure S13. In general, the perforation are parallel lines on the CEM membrane, with a gap in the range of 30 μm and a spacing of 3 mm. That is about 10% of the active surface area with open perforation. We did conduct a very brief study by increasing the perforation density, and we saw worse performance with more formic acid oxidation at the anode, as shown in the illustration figure below. Our understanding is that there are three different transport pathways for formic acid that is generated at the PCEM/AEM interfaces. The first is it will transport through the perforation and gets collected at the anode exhaust, the second is that it will meet with anode catalyst and then gets oxidized, the third is back diffuse to the cathode. When we increase the perforation area, there is a higher chance for formic acid to transport to the anode catalyst and gets oxidized. The perforation area or the perforation pore size should be large enough to let formic acid to come out, but not too large to cause too much anode formic acid oxidation.

Changes:

(Supplemental Figure S13)

(Line 342-350)

used as the anode catalyst layer ionomer. The CEM perforation was conducted by cutting parallel lines on the CEM membrane with a spacing of 3 mm. Details of the perforation process is illustrated in Figure S13 b and c. The perforation has a gap of 32.6 μm , confirmed by X-ray computed tomography, as shown in Figure S13 d and e. During cell assembly, the perforated catalyst coated CEM membrane was placed on a 25 cm² Toray paper (5 wt.% PTFE treated, Fuel Cell Store, USA). An AEM membrane (PiperION, Versogen, USA) was placed on top of the CEM and the cathode GDE on top of the AEM. A PTFE gasket was used for both anode and cathode with thicknesses to achieve an optimum GDE compression of 18%. Details of the cell

assembly process is shown in Figure S13 a.

2. The quantification and discussion of CO₂ crossover through the system should be included. It is unclear if CO₂ crossover has been included in the TEA model.

Response: Thank you for pointing this out. We have not yet quantified the CO₂ crossover in our system yet. The main challenge we are facing right now is there are two different CO₂ sources at the anode, first is the CO₂ crossover from the cathode to the anode, the second is CO₂ generated during formic acid oxidation. We find it is challenging to separate these two CO₂ sources using our current GC system.

But we can do a very brief estimation of the amount of CO₂ generated at the anode. The main reaction at the cathode is CO₂ reduction to formate ion. If we assume a CO₂ to Formic acid FE of 75%. CO₂ to CO FE of 10% and CO₂ to H₂ FE of 15%.

For current density I , the amount of CO₂ consumed at the cathode is $\frac{I}{2F}$, where F is the Faradaic constant.

Based on charge conservation, the amount of charge transport through the AEM membrane should also equal to I . We collected most of our formic acid from the anode, which means that HCOO⁻ is the main charge carrier in the AEM membrane. If we assume most of the other part of charge carrier is carbonate, (it is unlikely to be bicarbonate, because the cathode has a neutral or basic environment), then the amount

of CO₂ crossing over is $\frac{(1-75\%) \times I}{2F}$, which means 25% higher CO₂ consumption if we take into consideration of CO₂ crossover. Even if we assume all the other charge carrier through AEM is bicarbonate, the CO₂ crossed over should be $\frac{(1-75\%) \times I}{F}$, which would result in a 50% higher CO₂ consumption.

From our TEA analysis, the CO₂ cost is a very small portion of the total cost. Even if the CO₂ cost is increased by 25% to 50%, it won't affect the main conclusions of the TEA analysis.

Nevertheless, we agree that conducting more rigorous CO₂ crossover quantification analysis is of significant importance. We recently placed an order for a Time-of-flight Mass spectroscopy (TOF-MS) in our lab, and we may be able to conduct more systematic quantifications of CO₂ cross over in a follow-up study.

3. It is interesting that water and formic acid do not cause flooding in the catalyst layer of the anode. Why did the author use carbon paper with 5% PTFE content?

Response: Thanks for pointing this out. We use 5% PTFE content Toray Paper to create a hydrophobic layer and prevent flooding. It might sound a bit counter-intuitive, but supplying a small amount of water to the anode can alleviate the anode flooding issue. This is because formic acid has a much smaller contact angle compared to pure water, which would result in more flooding at higher formic acid concentrations. The formic acid transport is depicted in the figure below.

The 5% PTFE Toray paper creates a hydrophobic layer. The supplied DI will flow at the top of the Toray paper. High concentration formic acid will generate at the PCEM/AEM interface, and it will transport through certain channels through the paper (larger pores) and gets diluted and bring out of the anode and gets collated at the anode exhaust. If we don't have the DI water supplied to the anode, the low contact angle, high concentration formic acid has a higher chance to cause flooding at the anode.

4. Could the authors explain why the perforation was employed only on CEM and not on both the carbon paper and CEM? The latter configuration may enhance the diffusion of formic acid from the CEM layer and prevent further oxidation.

Response: Thank you for your question. The carbon paper that we are using is Toray paper. This kind of diffusion media that doesn't contain any MPL. The pores/ spacing between the carbon fibers have a size of ~100 um scale, which we believe is large enough and doesn't create significant capillary pressure for liquid to transport from the CEM/AEM interface to the outside. If Sigracet type carbon paper is used at the anode side, which contains a thick MPL layer, then maybe a perforation is necessary to enhance the anode formic acid transport.

Reviewer #3 (Remarks to the Author):

In this manuscript, a novel CO₂RR electrolyzer was designed by introduced a perforated cation exchange membrane (PECM) to facilitate the mass transport of formic acid generated at the membrane interface. The PECM cell exhibited good stability and high formic acid selectivity under high current. Authors also demonstrated the value of this technique through a techno-economic analysis. Though this novel structure of CO₂RR electrolyzer is of interest for the electrocatalytic community, the reviewer thinks the study and the presented data are not comprehensive. Some questions and concerns need to be addressed before the manuscript can be assessed for publication.

1, The control device using a cation exchange membrane should be constructed. And the issue of this kind of device also should be discussed.

Response: Thanks for the suggestion. We did tests using single CEM MEA configuration, similar to the work by Lee W, Kim Y E, Youn M H, et al. *Angewandte Chemie International Edition*, 2018, 57(23): 6883-6887. Results are added to the Supplemental Figure S1 e-g. When single CEM is used, cations like potassium, sodium or proton will migrate from the anode to the cathode, while it is hard for formate to

transport from the cathode to the anode through the CEM. Because of that, salt like potassium formate will accumulate and then get collected from the cathode. The initial performance of this system looks good, but the potassium formate accumulation at the cathode can lead to severe durability issues, like the salt will precipitate and block the cathode catalyst layer. After two hours of operation, significant amount of salt accumulation can be observed, especially at high current densities ($>100 \text{ mA/cm}^2$). Photo images of the GDE and flow field after two hours of operation at 200 mA/cm^2 is also included in the Supplemental Figure S12.

Changes:

(Line 112-115)

anolyte concentration is reduced to 0.1 M KOH, both cell voltage and formate oxidation (formate loss) increase (Figure S1d), illustrating the zero-sum tradeoff. The performance of using a MEA configuration and single CEM membrane is also investigated, with results shown in Figure S1 f,g. The formate FE collected from the cathode is $>60\%$ at 200 mA/cm^2 at beginning of test, but suffers from fast degradation within two hours due to cathode salt accumulation (Figure S12).

(Supplemental Figure S1)

(Supplemental Figure S12)

Figure S 8. Images of the (a) back of the cathode GDE and (b) flow field of the single CEM MEA after 2 hours of operation under 200 mA/cm^2 . Significant amount of salt accumulation can be observed.

2, Detailed information about the PCEM perforated cation exchange membrane, such as optical pictures and its preparation method, should be provided. The author should also provide physical pictures of each component of the equipment and mark the function of each component in the actual working equipment.

Response: Thanks for your suggestions. We've added a supplemental Figure S13 with details of the perforation process. We also updated the method section to include more descriptions of the perforation steps. As for components of the equipment, we put more details to the Supplemental Figure S5 to show each component of the equipment and their corresponding functions.

Changes:

(Supplemental Figure S13)

(Line 342-350)

used as the anode catalyst layer ionomer. The CEM perforation was conducted by cutting parallel lines on the CEM membrane with a spacing of 3 mm. Details of the perforation process is illustrated in Figure S13 b and c. The perforation has a gap of 32.6 μm , confirmed by X-ray computed tomography, as shown in Figure S13 d and e. During cell assembly, the perforated catalyst coated CEM membrane was placed on a 25 cm² Toray paper (5 wt.% PTFE treated, Fuel Cell Store, USA). An AEM membrane (PiperION, Versogen, USA) was placed on top of the

CEM and the cathode GDE on top of the AEM. A PTFE gasket was used for both anode and cathode with thicknesses to achieve an optimum GDE compression of 18%. Details of the cell assembly process is shown in Figure S13 a.

3, Authors should conduct a systematic investigation on the pore size and distribution of cation exchange membrane on the performance of PCEM.

Response: Thank you for your comments and suggestions. Our perforation size (or pore size) is 30 μm with a spacing of 3 mm for each parallel perforation lines, meaning that about 10% of the membrane area is void space. Our initial study here was to examine the impact of the perforation in a binary sense, just to check if it works or not in a forward bias bipolar membrane. More anecdotally, we did conduct a very brief study increasing both the perforation density and complexity, seeing worse performance and more formic acid oxidation at the anode, as shown in the illustration figure below. Our understanding is that there are three different transport pathways for formic acid that is generated at the PCEM/AEM interfaces. The first is it will transport through the perforation and gets collected at the anode exhaust, the second is that it will meet with anode catalyst and potentially become oxidized, the third is back diffusion to the cathode. When we increase the perforation area, there is a higher chance for formic acid to transport to the anode catalyst and gets oxidized. The perforation area or the perforation pore size should be large enough to let formic acid to come out, but not too large to cause too much anode formic acid oxidation.

But we agree that the parameters like the shape of the perforation, the arrangement of the perforation location, and perforation area are of great value for further investigation. We plan to do a more systematic investigation on that topic in a follow up study.

4. Figure S3 shows the durability of the system with an 80 mm AEM and perforated CEM at 60°C, What is the reason for the periodic fluctuation of the data in the figure?

Response: Thanks for pointing this out! We also noticed this very interesting cell voltage behavior. We think it may have something to do with anode Pt gets poisoned by trace amount of CO. For long-term stability runs, there is always a small amount of formic acid that gets oxidized at the anode.

It has been reported in literature that three different pathways for formic acid oxidation ((Gao W, Keith J A, Anton J, et al. JACS, 2010, 132(51): 18377-18385.)), as shown below, including a direct pathway that formic acid gets oxidized to CO₂, and an indirect pathway that formic acid first gets oxidized to CO. The second pathway will generate CO and slowly poison the Pt catalyst. As more Pt surface gets poisoned, the anode potential will increase. Once the anode potential reaches certain values, CO will get stripped away (or in other words, gets oxidized). In a MEA configuration, that CO stripping has a potential window of 0.4-0.6 V vs SHE. Once all CO get cleaned off from the Pt surface, the anode potential will drop, and this becomes a full cycle, leading to periodic fluctuation of the cell voltage.

(Gao W, Keith JA, Anton J, et al. JACS, 2010, 132(51): 18377-18385.)

(CO stripping curve at 100%RH for a MEA configuration on Pt surface)

5. How is formic acid effectively collected in actual devices.

Response: Thank you for your suggestions. I've modified the Supplemental Figure S5 to include more details of our experimental setup. As discussed in the main text, formic acid will be collected from the anode side as liquid form. I also added more descriptions in the main text about how it will be collected.

Changes:

(Supplemental Figure S5)

Figure S10. Detailed information about all the inputs and outputs of the cell during electrochemical cell testing. The anode is supplied with a mixture of 0.8 slpm H_2 at 100% RH and 10 mL/min DI water. The cathode is supplied with 2 slpm 100% CO_2 gas mixed with 2 mL/min 1 molar KOH. Formic acid is collected at the anode, and the CO and H_2 are collected from the cathode.

(Line 325-330)

to help remove formic acid at the anode. Detailed information about the cell inputs and outputs are shown in figure S5. The cathode effluent gas contains CO_2 and generated CO and H_2 , and it will pass through a condenser (low temperature heat exchanger with a temperature of 2°C) to get rid of water vapor. The remainder of the gas will be collected for gas chromatography analysis. The anode effluent will also go through the condenser to separate the liquid from the gas. The liquid effluent will be collected in a clean vial and then get analyzed using liquid chromatography to quantify the formic acid generated. The electrochemical testing was conducted using a Garmy Potentiostat

6. Is it possible to quantify the proportion of formic acid oxidation current in the total anodic current.

Response: Thanks for the suggestion. We have the formic acid oxidation partial current plotted on Supplemental Figure S11. It is obtained based on the charge and mass balance. The proportion of the formic acid oxidation partial current in the total current equals the value of the Faradaic efficiency labeled as FA loss.

Figure S 11. Partial current densities for different reactions when the cell is operated at 30, 60, and 80 °C with Pt/C anode and 80 μm AEM.

REVIEWERS' COMMENTS

Reviewer #1 (Remarks to the Author):

The authors have sufficiently addressed all the questions. The manuscript deserves to be published in the journal Nature Communications.

Reviewer #2 (Remarks to the Author):

The authors have addressed my concerns/comments.